# Hyperthermia: The Optimal Treatment to Overcome Radiation Resistant Hypoxia

**DOI:** 10.3390/cancers11010060

**Published:** 2019-01-09

**Authors:** Pernille B. Elming, Brita S. Sørensen, Arlene L. Oei, Nicolaas A.P. Franken, Johannes Crezee, Jens Overgaard, Michael R. Horsman

**Affiliations:** 1Department of Experimental Clinical Oncology, Aarhus University Hospital, DK-8000 Aarhus C, Denmark; pernille.elming@oncology.au.dk (P.B.E.); bsin@oncology.au.dk (B.S.S.); jens@oncology.au.dk (J.O.); 2Department of Radiation Oncology, Amsterdam University Medical Centers, University of Amsterdam, 1105AZ Amsterdam, The Netherlands; a.l.oei@amc.uva.nl (A.L.O.); n.a.franken@amc.uva.nl (N.A.P.F.); h.crezee@amc.uva.nl (J.C.)

**Keywords:** hyperthermia, radiation therapy, hypoxia

## Abstract

Regions of low oxygenation (hypoxia) are a characteristic feature of solid tumors, and cells existing in these regions are a major factor influencing radiation resistance as well as playing a significant role in malignant progression. Consequently, numerous pre-clinical and clinical attempts have been made to try and overcome this hypoxia. These approaches involve improving oxygen availability, radio-sensitizing or killing the hypoxic cells, or utilizing high LET (linear energy transfer) radiation leading to a lower OER (oxygen enhancement ratio). Interestingly, hyperthermia (heat treatments of 39–45 °C) induces many of these effects. Specifically, it increases blood flow thereby improving tissue oxygenation, radio-sensitizes via DNA repair inhibition, and can kill cells either directly or indirectly by causing vascular damage. Combining hyperthermia with low LET radiation can even result in anti-tumor effects equivalent to those seen with high LET. The various mechanisms depend on the time and sequence between radiation and hyperthermia, the heating temperature, and the time of heating. We will discuss the role these factors play in influencing the interaction between hyperthermia and radiation, and summarize the randomized clinical trials showing a benefit of such a combination as well as suggest the potential future clinical application of this combination.

## 1. Introduction

Hypoxia is a hallmark of solid tumors [1,2]. It occurs because the tumor neo-vasculature that develops from the host vascular supply via angiogenesis [3] is a primitive and chaotic system that is unable to meet the oxygen demands of the growing tumor mass [2,4]. As a consequence, cells that are distant from the blood vessels and at the limit of the diffusion distance of oxygen become chronically hypoxic [2,4]. Flow through tumor vessels is also unstable and periodically fluctuates and this can give rise to transient or acute hypoxia [2,4]. Regardless of the type of hypoxia, both pre-clinical and clinical studies show that its presence in tumors is a major factor influencing malignant progression and response to treatment, especially radiation therapy [2,5]. These observations have led to extensive pre-clinical and clinical attempts during the last 5 decades or so to try to specifically target this hypoxia and thereby improve patient outcome [6,7]. Examples of the approaches used are listed in Table 1. They include improving oxygen availability, increasing the radio-sensitivity of the hypoxic cells, killing the hypoxic population, or modifying the radiation treatment either by increasing the dose to the hypoxic areas (dose painting) or utilizing radiation of a higher LET (linear energy transfer) in which the oxygen enhancement ratio (OER) is reduced. It is interesting that hyperthermia (heat treatments of 39–45 °C) actually induces many of these effects and as such may have the potential to be one of the best agents for eliminating hypoxia.

The concept of using heat to treat cancer is actually not a new idea. In fact, it is probably one of the oldest documented treatments, since the Edwin Smith Surgical papyrus dating back to 3000 years B.C. describes a patient with a tumor in the breast treated with heat in the form of red-hot irons [8]. Early Greek (Hippocrates 400 B.C.) and Roman (Galen 200 A.D.) translations similarly recorded the use of heat treatments [9]. After the 17th Century, there were numerous reports of tumor regression in patients suffering with infectious diseases, which ultimately led to the application of fever-induced treatment with Coley’s toxin to control tumors at the end of the 19th Century [9]. Fever-induced treatments typically required temperatures of around 40 °C for several days. Later studies utilized primitive external heating techniques to achieve higher temperatures for shorter time-periods [9]. What is novel is that today we have new developments in technology (i.e., electromagnetic, ultrasound, infrared, and nano-technology based techniques) available to us [10,11]. We can thus select the appropriate technique for the specific tumor location and size, and with support from thermometry and quality assurance, we can now apply reproducible and uniform high quality hyperthermia treatments [11,12,13]. As a result, there have been numerous clinical studies showing the potential of hyperthermia to improve radiotherapy outcome [14,15,16,17]. In this review, we will discuss the different ways in which hyperthermia has been combined with radiation therapy, focusing on why this approach is selective for hypoxia, and suggest the clinical potential of this therapeutic combination to significantly improve patient outcome.

## 2. Combining Hyperthermia with Radiation

Tumor response to the combination of radiation and hyperthermia is dependent on the time interval between the two modalities, the heating temperature, and the time of heating [15,18,19], as illustrated in Figure 1. For the C3H mammary carcinoma model, shown in this example, the greatest enhancement of radiation response by heat occurs when both treatments are given simultaneously (Figure 1A). However, if any interval is introduced between the two modalities then the enhancement decreases with this decrease becoming greater as the time interval increases, eventually reaching a plateau as shown. A recent clinical study in patients with cervical cancer confirms the superior benefit of using a short rather than long interval between the radiation and hyperthermia treatments [20]. For the example shown in Figure 1A, the decreased response as the time interval increases is independent of whether heating is applied before or after irradiating, and this seems to be a general result for other tumor models [15]. Also shown in Figure 1A, is the effect of combining radiation and heat in a normal tissue. A simultaneous application of radiation with heat results in an enhanced effect that is identical to that seen in tumors. However, unlike the tumors, the drop-off is steeper when there is an interval and the final plateau reached is lower. For heat given after irradiating, this drop-off actually reaches a value where no enhancement occurs, but when heat is applied prior to irradiating, a residue enhancement remains. Again, the trends seen with the model shown in Figure 1A has been reported for other normal tissues [15,21].

The heating temperature and time of heating also influence the enhancement (Figure 1B). Generally, the higher the temperature and the longer the heating period, the greater the enhancement [15,18,19]. This is true regardless of the interval between the two modalities [15,18,19]. Although, as shown in Figure 1B, the degree of enhancement does depend on whether the radiation and heat are applied simultaneously or if there is an interval.

## 3. Radio-Sensitization by Hyperthermia

The consensus opinion is that the enhancement of radiation response by hyperthermia, when both treatments administered simultaneously, reflects radio-sensitization. This decreases as the interval between the two modalities increases ultimately disappearing when the interval is long enough (around 4 h in Figure 1A). The remaining enhancement seen as the plateau in Figure 1A is simply the result of heat killing hypoxic cells. Since the effects of a simultaneous treatment occurs in both the tumor and normal tissue, it would suggest that sensitization does not involve hypoxia. However, the normal tissue shown in Figure 1A is skin and that is known to be somewhat hypoxic [22,23] and found to have an increased radiation sensitivity when treated with classical nitro-aromatic hypoxic cell sensitizers [24]. Similar peak enhancements have been reported by others in skin and other normal tissues in which hypoxia is even absent [15,21], although in those studies no comparison was made with tumor response under similar conditions, so it is impossible to state whether the peak normal tissue response was equivalent to that for tumors or actually lower. What is interesting is that the radio-sensitization seen with hyperthermia administered in the temperature range of 40.5–42.5 °C at the same time as irradiating is as good as, or even better than, that found using nitro-aromatic drugs that specifically sensitize hypoxic cells to radiation (Figure 2). However, unlike the nitro-aromatic radio-sensitizers in which there is only a small effect in normal tissues [24] with hyperthermia one sees exactly the same effect in normal tissues and tumors [15,18,19], thus there is no therapeutic benefit for sensitization unless the heat and radiation treatments specifically target the tumor.

Interestingly, the enhancement of radiation response by hyperthermia can be further increased by simply combining the radiation and heat treatment with the radio-sensitizer misonidazole [25,26,27]. This additional benefit was seen with both a simultaneous and sequential radiation and heat treatment, and increased with temperature and drug dose [27]. Misonidazole specifically targets hypoxic cells [5,6] and if we assume that this is also true for hyperthermia, then it suggests that they may be acting on different hypoxic subpopulations. In fact, one study clearly showed that combining nicotinamide with hyperthermia was an effective approach to improve radiation response by reducing hypoxia [28] and that since nicotinamide only affected acute hypoxia the additional benefit of heat must have been through a reduction in chronic hypoxia.

Two different mechanisms are believed to account for the radio-sensitization by hyperthermia. The first involves an improvement in oxygen delivery to the tissue. Over two decades ago, it was proposed that the clinical benefit of hyperthermia was the result of mild hyperthermia temperatures (less than 42 °C) improving radiation response by increasing tumor oxygenation status, resulting in a corresponding decrease in radiation resistant hypoxia [29]. Pre-clinical data clearly shows that mild heat temperatures can improve tumor oxygenation [30,31,32,33,34] and this is most likely the result of changes in tumor blood flow [32,33,34,35], perhaps mediated through a decrease in interstitial fluid pressure [33,34] causing an increase in perfused vessels [34,36]. Interestingly, the improvement in tumor oxygenation and radiation response observed with mild temperature hyperthermia, could be further increased if animals were allowed to breathe carbogen gas immediately after heating and during the radiation period [37]. Studies have generally shown that at mild heat temperatures tumor blood flow and oxygenation status temporarily increases during the heating period but returns to normal values afterwards [31,32,35,38]. However, the radio-sensitization by heat increases with temperature and while temperatures above 42 °C might produce a very transient increase in oxygenation during the heating period, immediately after heating there is a rapid induction of vascular damage that would be expected to significantly increase the level of hypoxia and thus radio-protect tumors [32,35,38]. Cellular oxygen utilization may play a role here. This actually increases at low temperatures but decreases at high temperatures [32]. However, the increase at low temperatures of around 41 °C is only transient and with time will actually drop [39]. Any decrease in oxygen consumption, if it occurs in tumors, can actually increase the diffusion distance of oxygen, thus improving oxygen availability to hypoxic regions and so decreasing the level of hypoxia [40,41], even at the high temperatures. The situation is made even more complicated by studies from one group reporting improved tumor oxygenation that lasted for 1–2 days after heating at mild temperatures [30,42,43,44,45], an effect that is difficult to explain. Clinical studies, in which the oxygenation status of soft tissue sarcomas [46] or locally advanced breast cancer [47] was measured with oxygen electrodes after heating, also seemed to indicate that mild temperature hyperthermia improved tumor oxygenation. However, in the sarcoma study the improvement in oxygenation correlated with the degree of necrosis found in resected specimens [46], suggesting that the apparent oxygenation effects simply reflected the damage caused by higher temperatures. Indeed, an additional pre-clinical study in which tumors treated with high thermal temperatures, known to induce physiological changes that cause a significant decrease in tumor perfusion and oxygenation [1], resulted in substantial tumor control [48] while actually reporting apparent improvements in tumor oxygenation 1–2 days after applying the heat. This suggests that the apparent improvements in oxygenation were not the result of improved oxygen delivery. Clearly, the role of heat-induced effects of tumor oxygenation status accounting for radio-sensitization is somewhat controversial and unclear. What is clear from rapid-mix studies is that the improved oxygen levels are only beneficial if they are present at the time of irradiation or within a few milliseconds after irradiating [49,50]. Yet the enhancement of radiation by heat is the same whether the heat is applied long before or after irradiating (Figure 1).

An alternative explanation for radio-sensitization by hyperthermia involves potential effects on radiation-induced DNA damage repair. Ionizing radiation causes different DNA lesions that include base damage, single strand breaks (SSBs) and double strand breaks (DSBs), the latter being produced either directly by an ionizing event or indirectly when SSBs are produced close to each other on both DNA strands [51] or indirectly when problems occur during DNA replication [52,53]. DNA DSBs are potentially the most toxic DNA lesions to cancer cells, repairable by two major pathways, non-homologous end joining (NHEJ) or homologous recombination (HR) [54,55]. NHEJ rejoins the broken ends of the DNA without the need for homology or a repair template and is active through the cell cycle [56,57], whereas HR does require a template thus is only active during the S and G2 phases of the cell cycle [56,58]. When a DSB occurs, kinases initially recognize the DNA break, and accumulate together with other kinases around the location of the break and attract other DNA repair proteins to repair the break [58]. During this process, histone H2AX (γ-H2AX) is phosphorylated; which explains its use as a marker for the induction and repair of DSBs [59]. Many studies demonstrated increased levels of γ-H2AX after combined treatment with ionizing radiation and mild hyperthermia as compared with ionizing radiation alone at 24 h after treatment, indicating that the number of residual DSBs was increased [60,61]. Moreover, after heat treatment, decreased levels are reported of BP53 and Rad51; these proteins are involved in recruiting other repair proteins of NHEJ and HR to the DNA break ends, suggesting that hyperthermia interferes with both DNA repair mechanisms [62]. How the DNA repair pathways are affected is only partially known. Multiple groups reported that hyperthermia affects NHEJ pathway-specific proteins. Hyperthermia is suggested to affect NHEJ by heat-mediated inactivation of Ku, decreased activity of DNA-PK and decreased levels of KU70, KU80, and Ligase IV [62]. Hyperthermia has also been found to temporarily degrade the BRCA2 protein and reduce BRCA1, thereby inhibiting the homologous recombination DNA repair pathway [63]. Interestingly, as a result of the poor supply of oxygen and nutrients, hypoxic cells often exist in a quiescent state [64]; these quiescent cells are less sensitive to ionizing radiation because they have the time to repair the DNA properly, as well as the known resistance due to the lower level of oxygenation. Hyperthermia can push cells out of this quiescent state and make them more susceptible to ionizing radiation [65]. 

## 4. Hyperthermia as a Cytotoxic Agent

Hyperthermia can also kill hypoxic cells either directly or indirectly. Direct cell killing is strongly dependent on the heating temperature and the time of heating; the higher the temperature and the longer the heating period, the greater the effect [66,67,68]. Typically, temperatures below 42 °C actually have little effect on cell killing, unless long exposure times are utilized [66,67,68], certainly longer than a typical heating period of 1 h when combined with radiation [14,15,16,17]. The killing seen with temperatures above 42 °C increases significantly if the cells are maintained under conditions of oxygen deprivation and/or low pH [69,70]. Such adverse micro-environmental conditions are those typically found in hypoxic tumor regions [1,2] and the ability of heat to actually kill hypoxic cells in tumors has been demonstrated [28,71]. The Overgaard study actually suggested that this heat-killing effect was primarily in chronically hypoxic rather than acutely hypoxic cells [71]. This seemed to be confirmed in the other study [28] and is supported by in vitro data, since long periods of hypoxic exposure were necessary to obtain cell killing [69,70]. In tumors, it is also the chronically hypoxic cells, rather than acutely hypoxic, that will be more likely associated with nutrient deprived conditions that also give rise to heat sensitive low pH. Preferential killing of radio-resistant hypoxic cells probably explains the plateau effect seen when heat and radiation are separated by greater than 4 h (Figure 1A). Analysis of the time-temperature cell survival curves also suggest that the slopes of the curves were very different above or below 42.5 °C [66,67,68], either reflecting different cell killing mechanisms or that with the longer heating times the cells developed resistance to heat, often referred to as thermo-tolerance [72]. Interestingly, the time-temperature response curves for tumors in vivo also shows different slopes above and below 42.5 °C [73] suggesting similar thermo-tolerance mechanisms as in vitro. However, additional studies suggest that thermo-tolerance in vivo may also be vascular mediated resulting from the induction of vessel normalization [74], a process that involves a decrease in micro-vessel density and increase in pericyte coverage [75], and causes a decrease in tumor hypoxia [76].

Heat kills cells by a variety of mechanisms, including necrosis, apoptosis and modes related to mitotic catastrophe [77,78,79,80]. The biological effects of heating cells include chromosomal aberrations, mitotic dysfunction, cytoskeletal damage, changes in membrane fluidity and transport, and metabolic changes [81]. But the most likely rate limiting step for killing by hyperthermia is protein denaturation since this in a similar time-temperature relationship as for cell killing, especially at temperatures of 42.5 °C and above, although some effects do occur with long heating times at lower temperatures [82]. At temperatures around 43 °C and below apoptosis appears to predominate with necrosis seen at higher temperatures [77,80], but whether these effects are mediated via denaturation of proteins associated with the cytosol, membrane, or nucleus, is unclear [77,79]. However, the fact that cell killing is substantially increased if cells are heated under low pH conditions [69,70] would seem to support the cell membrane as the primary target. Measurement of intracellular pH (pHi) and extracellular pH (pHe) show that cells can maintain a neutral pHi even when pHe is acidic [1]). Increasing extracellular acidity would put more stress on the membrane pumps responsible for maintaining neutral intracellular pH and thus be more susceptible to heat damage.

Whatever the mechanism, one can significantly increase heat killing in vivo using agents that decrease tumor blood flow and, thus, increase the adverse environmental conditions, especially hypoxia, within tumors. This has been achieved using physiological modifiers, such as hydralazine, sodium nitroprusside, or glucose [83]; the effects are often transient and hard to predict, yet have been shown to enhance tumor response to heat [83]. More consistent, longer-maintained changes are seen with so-called vascular disrupting agents (VDAs) that damage the established tumor vascular supply [83,84,85]. Many of these VDAs, including tumor necrosis factor, chemotherapeutic drugs (e.g., arsenic trioxide and vinblastine), flavonoid compounds (e.g., flavone acetic acid and vandremycin) and tubulin-binding agents (e.g., combretastatin and its analog OXi4503), have been combined with hyperthermia to enhance the anti-tumor response [85]. Additional benefits were observed when VDAs and hyperthermia were combined with radiation [15,85]. Moreover, such combinations were extremely effective when using mild temperature heat treatments; the radiation induced tumor control reported after systemic treatment of mice with VDAs and local tumor heating at 41.5 °C was as good as, if not better than, that seen with 43 °C alone [15,85].

Hyperthermia itself can also induce vascular damage and, as a result, will kill tumor cells indirectly. Although higher heat temperatures may transiently increase tumor blood flow during the heating period, immediately after the cessation of heating a rapid decrease in tumor blood flow is seen that is often prolonged (Figure 3), although the effects are tumor and temperature dependent. This rapid and prolonged vascular collapse following heating is similar to that seen after treatment with the drug based VDAs [84,85]. For such drugs, the target is the vascular endothelial cells [86], which when damaged undergo a rapid shape change and eventually undergo apoptosis [86]. The initial effects will cause an increase in vessel permeability and as fluid leaks out of the tumor vessels it will cause a rise in interstitial fluid pressure that then collapses the vessels [86]. This is likely to be transient, but would initiate a number of other effects such as reduced blood flow, increased blood viscosity and red cell stacking, and these would cause coagulation and thus responsible for the prolonged effect. The result of the vascular shutdown will deprive tumor cells downstream of the blockage of essential oxygen and nutrients, resulting in rapid and widespread tumor necrosis and ischemia [84,87]. The cells that die first are likely to be those that already exist under deprived conditions, especially hypoxic cells. Interestingly, despite the massive necrosis induced by VDAs their potential as stand-alone therapeutic agents is limited and for their full clinical potential to be realized they need to be combined with more conventional therapies, especially radiotherapy [84,87]. This is exactly the same situation with hyperthermia and is another strong argument for combining heat and radiation [15].

## 5. Hyperthermia and Alternative Radiation Approaches

The only category in Table 1 where hyperthermia is not listed is the use of novel radiation-based approaches to overcome hypoxia. These involve either increasing the radiation dose to the hypoxic areas identified from imaging analysis (dose painting) or utilizing high LET radiation (i.e., carbon ions) leading to a lower OER. Dose painting approaches [89] have generally been limited because they typically involve positron emission tomography (PET) based imaging technology that fails to represent the hypoxic distributions within tumors [4,7], thus non-hypoxic areas could receive a higher dose of radiation, while regions with hypoxia are missed. However, a recent pre-clinical study [90] suggests that oxygen images obtained using electron paramagnetic resonance could identify hypoxic tumor regions to which radiation was boosted to improve local tumor response. On the other hand, effective application of hyperthermia should target all hypoxic cells regardless of where they are located, thus making dose painting redundant. 

There is clear pre-clinical evidence that as LET increases the OER decreases [91,92]. At sufficiently high LET, as seen with heavy ions such as carbon ions, the OER is extremely low so hypoxia becomes less of an issue, but in a clinically obtainable LET range hypoxia is not entirely eliminated. The advantage of using heavy ions is the improved dose distribution to the tumor thus reducing normal tissue complications. Unfortunately, there are currently only 11 heavy ion facilities in the world (almost half located in Japan) [93]. Interestingly, proton facilities are for more common (there are currently some 70 facilities operational around the world with other facilities in development) [93] and even though protons have a LET that is significantly lower than carbon ions, thus hypoxia is still a significant problem, protons and carbon ions actually share similar physical advantages. It has been suggested that the combination of hyperthermia and protons may mimic carbon ion therapy [94] and as a result a clinical trial (HYPOSAR) of hyperthermia and protons in sarcoma patients is already underway [94]. This trial will involve applying heat temperatures of 41.5–42.5 °C for 60 min some 90–150 min prior to the first of five daily irradiations given each week over a seven-week period. However, this is a new concept based on theory and limited in vitro cell survival data [95], so whether the planned tumor temperatures and time interval will be as effective as carbon ions is unclear. Clearly, detailed in vivo studies looking at the combination of hyperthermia and protons are required, but studies with photons and hyperthermia may give some idea of the potential success of this approach. From single dose studies in murine tumors, we know that carbon ions are 1.4–2.4 times more effective than photons [96]. Similar enhancement ratios are shown in Figure 1A with heating at 42.5 °C for 60 min at time intervals ranging from 0 to 240 min, and has also been seen with other tumor models [15], supporting the use of this temperature and proposed time intervals in the clinical study. At a lower temperature of 41.5 °C, reduced enhancement ratios have been reported [97], decreasing from 1.7 with a simultaneous heat and radiation treatment to 1.2 with an interval of only 120 min. This suggests that for the selected time intervals between heating and irradiating, the lower temperature of 41.5 °C may be close to the limit of benefit. However, since protons have an increased relative biological effectiveness (RBE) compared to photons [98], with a RBE of 1.1 currently generically applied for clinical use, the effect of protons at 41.5 °C may actually be somewhat higher than predicted from photon studies. Obviously, pre-clinical testing of protons and hyperthermia, using a range of temperatures and time intervals is required to support the planned clinical studies.

Interestingly, there appear to be radiobiological differences between photon and proton irradiation [99,100]. Although the initial number of DNA damage foci increases with LET [101,102], the actual number are comparable between photons and protons when irradiating with protons using therapeutic beams with relatively low LET [103,104]. What has been found to be different is the residual number of unrepaired DSBs [102,104,105,106] suggesting that the repair processes are different following photon and proton irradiation, which may be part of the RBE of 1.1 for proton irradiation. It has been suggested that DSBs induced by high LET radiation, are preferentially repaired by HR [107,108], possibly because the short DNA fragments induced by the high-LET clustered DNA damage are unable to bind the Ku heterodimer. Several studies suggest this may also be the situation with protons [103,109,110], although data from at least one study has indicated that NHEJ also plays an important role in repairing DSBs induced by protons [108]. Interestingly, an in vitro study using cells with normal repair capacity, or deficient in either NHEJ or HR, investigated the effect of heating (42.5 °C for 1 h) immediately after irradiating with X-rays or protons [111]. The authors reported an enhancement of radiation-induced cell killing in the normal or NHEJ deficient cells, but not in the HR deficient, thus if HR is the principal repair mechanism after proton irradiation it adds further support to the potential combination of heat and proton irradiation. The authors also showed the same effect for carbon ions. This taken together with other studies showing the benefit of combining hyperthermia with high LET radiation [112,113,114,115], and the fact that hypoxia is less of an issue with high LET radiation [91,92], suggest that the combination of high LET and hyperthermia may be the ultimate approach for totally eliminating tumor hypoxia.

## 6. Clinical Relevance

Hyperthermia has been combined with radiation in a large number of clinical trials [14,15,16,17]. A meta-analysis of trials in which the patients were randomized to receive radiation alone or radiation and heat is summarized in Figure 4 [116,117,118,119,120,121,122,123,124,125,126,127,128,129]. Most of the studies involved conventional fractionated radiation therapy although one study with cervix cancer patients applied brachytherapy [119]. The planned heat treatment was typically 42–43 °C, for a period of 30–60 min, although in the brachytherapy study the aim was for a slightly lower temperature of around 41 °C [119]. Hyperthermia was applied either once or twice weekly after radiotherapy in all the studies, but with a time interval between the radiation and heat that varied from a few minutes up to 4 h. The endpoint for these studies was complete response/local tumor control, which is understandable because the heat and radiation were applied locally to the primary tumor, with sometimes a few local lymph nodes being treated. Some of the studies also reported survival data [117,118,119,122,123] but the significance of this when using local treatments is questionable. When considering tumor site, the analysis showed a significant improvement in local tumor control when hyperthermia was combined with radiation for all sites except lung. This lung study was an International Atomic Energy Agency trial in non-small cell lung cancer patients [124] and the failure to show any benefit may simply reflect poor quality control resulting from poor communication infrastructure in developing countries rather than any biological basis. Even including the results of this trial in the overall analysis, there was still a highly significant better response for radiation and heat compared to radiation alone.

Today, patients typically receive chemotherapy with radiation as part of the standard treatment, so one could question the validity of combining heat and radiation. Several studies have actually shown that including hyperthermia in the radiation/chemotherapy schedule can improve outcome [14,16]. Interestingly, a recent study in which patients with locally advanced cervical cancer were randomized to radiotherapy + hyperthermia or radiotherapy + cisplatin, reported comparable outcome and toxicity between the two treatment arms [131], suggesting hyperthermia might have a role to play as an alternative treatment if chemotherapy tolerance issues arise. Furthermore, hypoxia is often a source of resistance to many clinically used chemotherapeutic agents. This can be a consequence of hypoxia per se, but could also be due to hypoxic cells being distant from blood vessels; thus, creating a drug delivery problem, or because hypoxic cells are generally non-cycling and exist at low pH, both of which can influence drug activity [2,5]. Thus, including hyperthermia in a treatment schedule involving radiation and chemotherapy would seem to be a logical choice. 

## 7. Future Perspectives

The future of hyperthermia involves technical developments and improved approaches for applying current methodology that allows for better and homogeneous tumor heating or being able to reduce the interval between the heat and radiation to the point where a truly simultaneous radiation and heat treatment occur. However, there are a number of biological issues, which have recently become hot topics, in which hypoxia and hyperthermia may play critical roles [132]. The first of these involves immune response. Numerous reviews have addressed the issue of immune modulatory effects of hyperthermia [133,134,135] and different mechanisms have been suggested that involve both the innate and adaptive immune system. When heat is applied to tumor cells they respond by producing heat shock proteins (HSPs) which when become extracellular act as danger signals for the adaptive and innate immune systems [136]. These released HSPs activate NK cells and antigen presenting cells (APCs) and thereby increase the cytotoxic T-cell response [135]. Heat itself can also cause cellular damage and this will lead to the release of damage-associated molecular patterns (DAMPs) which basically has the same effect as the HSPs [133]. There is evidence that immune cells, such as NK cells, CD8+ T-cells, and dendritic cells already in the tumor also become activated when heated [135]. Finally, immune cell trafficking into the tumor can be improved as a result of the vascular effects of heating [134]. More recent studies have shown that the combination of hyperthermia and radiation can further enhance the immune response, probably by the induction of a greater proportion of immunogenic cell death in the tumor [137,138]. The significance of hypoxia in this issue comes from the finding that hypoxia in tumors can have a negative effect on immunogenicity by altering the function of immune cells and/or increasing resistance of tumor cells to the cytolytic activity of immune effectors [139,140]. The elimination of hypoxia by hyperthermia adds to its already established immune modulatory effects. These effects on immune response will not only impact the primary tumor, but should also induce an abscopal effect. Indeed, anti-tumor activity in contralateral tumors that are not actually heated has been reported [141,142], but whether the same applies to truly metastatic disease is not known. Clearly, additional studies into the role of hypoxia in the immune response and its targeting by hyperthermia should be undertaken. Furthermore, these studies should be extended to consider what happens when heat and radiation are combined. 

Successful cancer therapy requires targeting both the primary tumor and metastases. The cells within a tumor generally consist of non-stem cells that have a limited proliferative capacity but constitute the bulk of the cancer cells and a subset of cancer stem cells (CSCs) that can both expand the CSC population and differentiate into the various tumor cell populations [143,144]. CSCs are not only important for the growth of the primary tumor, they also play a major role in influencing metastatic spread [145,146]. Evidence exists that such CSCs are also an important factor in influencing response to treatment [144,146]. This is especially true for radiation in which studies have shown that the higher the proportion of CSCs, the greater the level of radiation resistance [147,148]. Part of this resistance may be due to intrinsic factors, such as greater repair capacity, and protection from reactive oxygen species-induced damage [149,150]. However, it is now becoming clear that additional extrinsic factors can influence resistance, especially the micro-environmental parameter of hypoxia [144,145,146]. Although an association between CSCs and hypoxia has been identified, the exact details of this relationship are unclear. It is not known whether CSCs and hypoxia simply co-exist, or whether hypoxia causes recruitment of non-stem cells in to the CSC pool. There is also the issue of how CSCs and hypoxia influence treatment resistance and malignant progression. Studies have reported that CSCs are more radiation resistant than non-stem cells [149], and hypoxia is definitely a critical factor in influencing resistance to radiation [2,5]. However, hypoxia may also increase radiation resistance by preventing cell differentiation and thus maintaining tumor cells in a more resistant undifferentiated “stem-cell-like” state [146,151].

It is also not clear as to how CSCs and hypoxia can influence malignant progression. Undifferentiated cells are more malignant [146,151], and with the hypoxic microenvironment contributing to the undifferentiated state of CSCs, this could partially explain how hypoxia and CSCs increase metastatic spread. Proteases, especially cathepsins (CTSs), are functionally involved in cancer progression including tumor invasion and metastases [152]. CTS expression, especially for CTS-L and K, have been associated with CSCs [153,154]. Such expression is also elevated under hypoxic conditions [155]. Clearly, targeting hypoxia using hyperthermia could be a novel approach for dealing with CSCs and thereby influencing both local response to radiation as well as malignant progression [156].

## 8. Conclusions

Figure 5 summarizes the critical issues in this review. Hypoxia is a characteristic feature of solid tumors that is a clinically relevant problem because it is a major resistance factor for conventional therapy, especially radiotherapy, and plays a significant role in malignant progression. Effectively decreasing hypoxia would clearly improve response to therapy and reduce the likelihood of metastatic spread. Despite numerous pre-clinical and clinical studies since the 1970s, only one agent has been established as a treatment option against hypoxia, and that is the radio-sensitizer nimorazole and only for treating head and neck patients with radiotherapy in Denmark. Since hyperthermia can effectively target hypoxia via a variety of different mechanisms, and has been shown to improve radiotherapy treatment in a number of tumor sites, it would suggest the application of heat to combat hypoxia should be adopted on a much wider basis, and even established as part of standard cancer therapy.

## Figures and Tables

**Figure 1 cancers-11-00060-f001:**
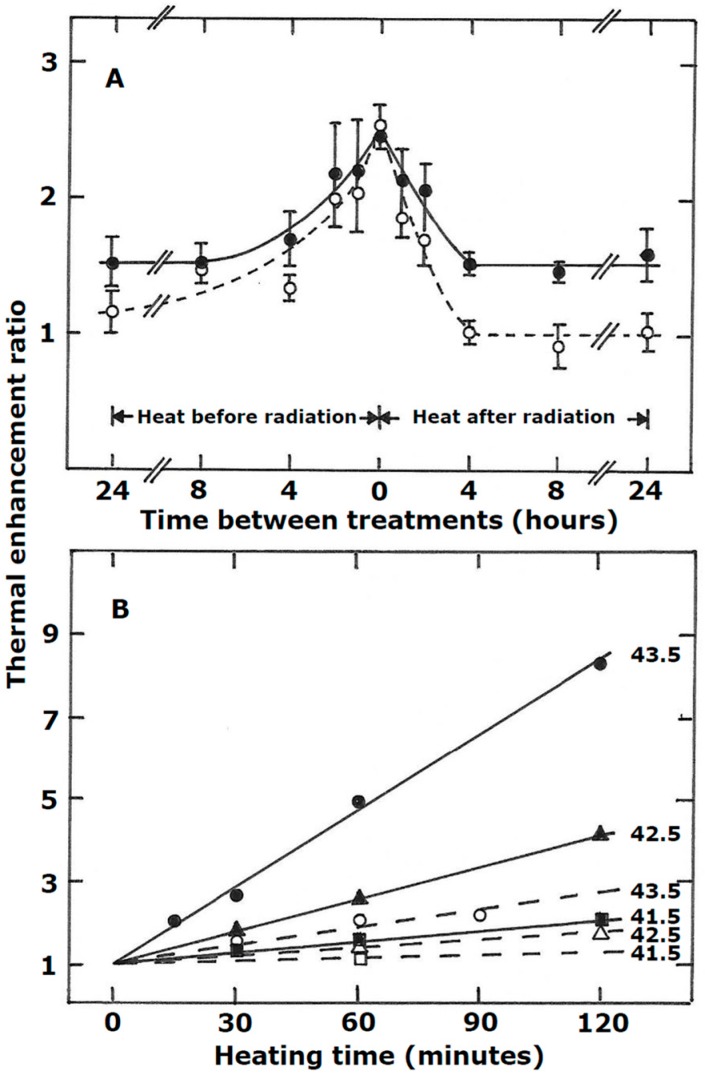
(**A**) Influence of time interval and sequence between radiation and hyperthermia (42.5 °C; 60 min) on tumor control in C3H mammary carcinomas (●) or moist desquamation in normal skin (○). (**B**) Effect of heating time and temperature on the thermal enhancement ratio (TER) for tumor control in a C3H mammary carcinoma when radiation and hyperthermia were given either simultaneously (solid symbols) or tumors irradiated and then heated 4 h later (open symbols); the heat temperatures are indicated. For both figures the TERs were determined from full radiation dose-response curves and represent the ratio of the radiation dose for radiation alone to that for radiation + heat to produce a response in 50% of animals. (Modified from [18,19]).

**Figure 2 cancers-11-00060-f002:**
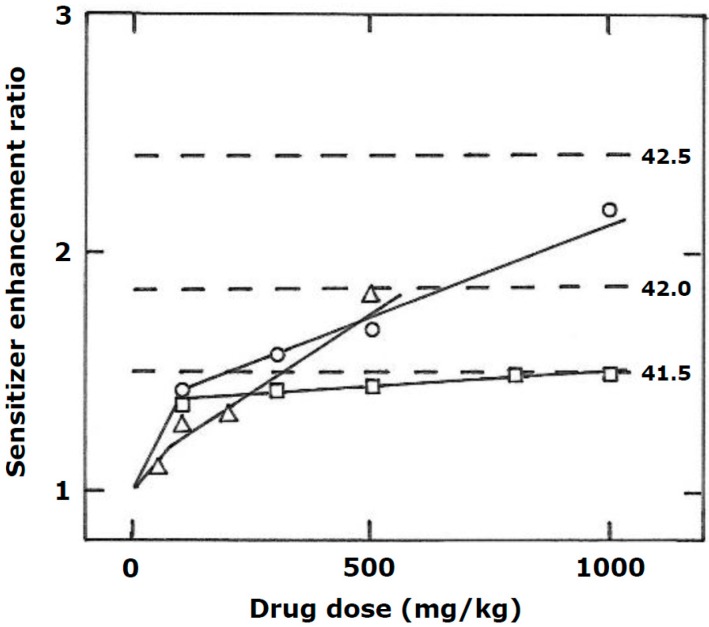
The radio-sensitizing effect of nitro-aromatic drugs and hyperthermia in a C3H mammary carcinoma. The sensitizer enhancement ratios (SERs) were calculated from full radiation dose-response curves of tumor control and represent the ratio of the radiation dose for radiation alone to that for radiation + sensitizer to produce a response in 50% of animals. The drug treatments were misonidazole (○), nimorazole (□), and doranidazole (△), with different drug doses administered as a single intraperitoneal (misonidazole and nimorazole) or intravenous (doranidazole) injection 30 min prior to irradiating (Modified from [24]). The dashed lines represent the SER levels when tumors were irradiated in the middle of a 60-min heating period at the indicated temperatures and are taken from Figure 1B.

**Figure 3 cancers-11-00060-f003:**
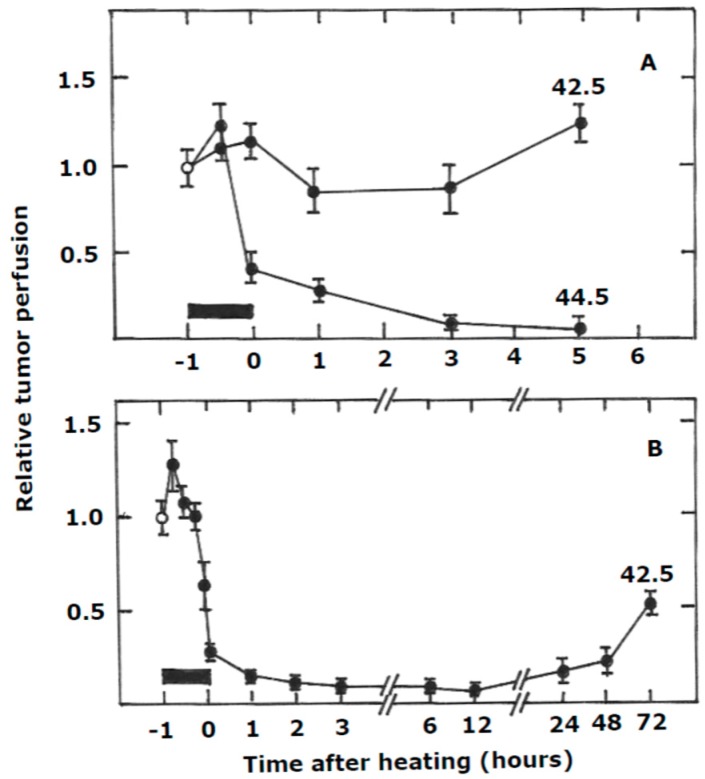
Effect of heating on perfusion in RIF-1 fibrosarcomas (**A**) or C3H mammary carcinomas (**B**). Tumors were heated for 1 h (shown by the black bars) at the indicated temperatures and blood perfusion in the tumors measured at different times before, during, or after heating by intravenously injecting radioactive rubidium chloride; tumors were excised 90–120 min later and tracer uptake measured on a gamma counter. Points are means (±1 S.E.) with the pre-treatment control values shown by the open symbols (Modified from [38,88]).

**Figure 4 cancers-11-00060-f004:**
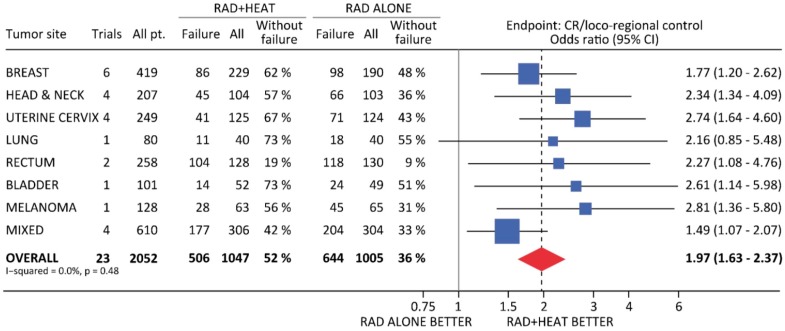
Meta-analysis of all trials in which patients were randomized to receive radiation alone (RAD) or radiation + hyperthermia (RAD + HEAT). The endpoint in each trial was complete response (CR) measured by loco-regional control and shows the calculated Odds ratio with 95% confidence intervals (95% CI). Data from [116,117,118,119,120,121,122,123,124,125,126,127,128,129] and observations from Overgaard, J. [130].

**Figure 5 cancers-11-00060-f005:**
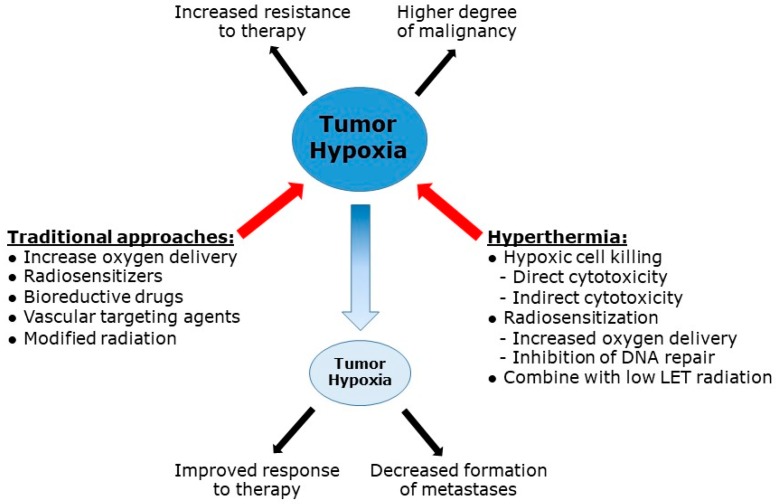
Hypoxia in tumors causes resistance to conventional treatments (i.e., radiotherapy of chemotherapy) and enhanced malignant progression (i.e., more aggressive growth of primary tumors or metastatic spread). Attempts to decrease hypoxia, and thereby improve tumor response to therapy as well as decrease the formation of metastases, have utilized a variety of different “traditional” approaches, as listed in Table 1. Hyperthermia can also decrease tumor hypoxia by a variety of mechanisms equivalent to all those seen with the more traditional methods.

**Table 1 cancers-11-00060-t001:** Approaches for dealing with hypoxia.

**Increasing oxygen delivery** - High oxygen content gas breathing (e.g., hyperbaric oxygen, carbogen) - Altering hemoglobin (e.g., transfusion, erythropoietin, perflurochemical emulsions) - Reducing fluctuations in flow (e.g., nicotinamide, pentoxifylline) - Decreasing oxygen consumption (e.g., metformin, phenformin) - Increasing blood flow (e.g., *hyperthermia*)
**Radio-sensitizing hypoxic cells** - Nitro-aromatic sensitizers (e.g., misonidazole, nimorazole, etanidazole, doranidazole) - *Hyperthermia*
**Preferentially killing hypoxic cells** - *Hyperthermia* - Bioreductive drugs (e.g., tirapazamine, banoxantrone, PR-104, evofosfamide)
**Vascular targeting therapies** - Angiogenesis inhibitors (e.g., avastin, DC101, tyrosine kinase inhibitors) - Vascular disruptive drugs (e.g., combretastatin, OXi4503, vademezan, *hyperthermia*)
**Radiation based approaches** - Dose painting - High LET (linear energy transfer) radiation

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
