# Peer review of "Hyperthermia: The Optimal Treatment to Overcome Radiation Resistant Hypoxia"

_cancers, 2019, doi:10.3390/cancers11010060_

Round 1

Reviewer 1 Report

The is a very nice summary of the various research efforts that have proven the value of thermal medicine to affect hypoxia in solid tumors and increase response to radiation therapy in the process.   I would suggest only a few additional considerations for the authors to potentially lay out an even broader and more comprehensive summary of the work that has supported the ideas presented here.  Namely, there are a couple papers that could be cited to add to the discussion/debate on page 5.  These have also demonstrated longer term changes in tumor blood flow, turmor interstitial pressure and oxygenation after mild heating and/or the addition of high oxygen content breathing.

Please consider

Mild elevation of body temperature reduces tumor interstitial fluid pressure and hypoxia and enhances efficacy of radiotherapy in murine tumor models.

Sen A, Capitano ML, Spernyak JA, Schueckler JT, Thomas S, Singh AK, Evans SS, Hylander BL, Repasky EA.

Cancer Res. 2011 Jun 1;71(11):3872-80. doi: 10.1158/0008-5472.CAN-10-4482. Epub 2011 Apr 21  PMID:  21512134

Select item 212

Tumour thermotolerance, a physiological phenomenon involving vessel normalisation.

Dings RP, Loren ML, Zhang Y, Mikkelson S, Mayo KH, Corry P, Griffin RJ.

Int J Hyperthermia. 2011;27(1):42-52. doi: 10.3109/02656736.2010.510495. Epub 2011 Jan 4.  PMID:  21204622

Mild temperature hyperthermia combined with carbogen breathing increases tumor partial pressure of oxygen (pO2) and radiosensitivity.

Griffin RJ, Okajima K, Barrios B, Song CW.

Cancer Res. 1996 Dec 15;56(24):5590-3.  PMID:  8971160

Related to this point, there is a high degree of self-citing by the authors.  The work is very good and relevant, however, it would serve the purposes of the review better if some other papers from different groups with similar results could be inserted to reduce this tendency.

Overall, this is seen as an important and well written review which will be of great value to the wider field of cancer research and treatment and should be published.

Author Response

Reviewer 1

Comments and Suggestions for Authors

The is a very nice summary of the various research efforts that have proven the value of thermal medicine to affect hypoxia in solid tumors and increase response to radiation therapy in the process.   I would suggest only a few additional considerations for the authors to potentially lay out an even broader and more comprehensive summary of the work that has supported the ideas presented here.  Namely, there are a couple papers that could be cited to add to the discussion/debate on page 5.  These have also demonstrated longer term changes in tumor blood flow, turmor interstitial pressure and oxygenation after mild heating and/or the addition of high oxygen content breathing.

Please consider

Mild elevation of body temperature reduces tumor interstitial fluid pressure and hypoxia and enhances efficacy of radiotherapy in murine tumor models.

Sen A, Capitano ML, Spernyak JA, Schueckler JT, Thomas S, Singh AK, Evans SS, Hylander BL, Repasky EA.

Cancer Res. 2011 Jun 1;71(11):3872-80. doi: 10.1158/0008-5472.CAN-10-4482. Epub 2011 Apr 21  PMID:  21512134

Select item 212

Tumour thermotolerance, a physiological phenomenon involving vessel normalisation.

Dings RP, Loren ML, Zhang Y, Mikkelson S, Mayo KH, Corry P, Griffin RJ.

Int J Hyperthermia. 2011;27(1):42-52. doi: 10.3109/02656736.2010.510495. Epub 2011 Jan 4.  PMID:  21204622

Mild temperature hyperthermia combined with carbogen breathing increases tumor partial pressure of oxygen (pO2) and radiosensitivity.

Griffin RJ, Okajima K, Barrios B, Song CW.

Cancer Res. 1996 Dec 15;56(24):5590-3.  PMID:  8971160

Related to this point, there is a high degree of self-citing by the authors.  The work is very good and relevant, however, it would serve the purposes of the review better if some other papers from different groups with similar results could be inserted to reduce this tendency.

Overall, this is seen as an important and well written review which will be of great value to the wider field of cancer research and treatment and should be published.

Response: We thank the reviewer for the very positive comments and for reminding us of the important work by others that we had forgotten to include. As a result, we have now added the suggested three references (Sen et al., 2011 now number 33; Dings et al., 2011, now number 74, and Griffin et al., 1996, now number 37) and modified the comments on page 5 (lines 163-171) and page 6 (lines 246-249) in line with the inclusion of these references.

We agree that we may have included too many self-citations. To remedy this we have removed our references number 8, 13, 20, 28, 35, 69, and 79 and added new references 8, 13, 17, 30, 34, 36, 72, 73, 75, 76, 89, and 130 (as well as 33, 37 and 74 listed earlier) from other groups. It was also necessary to change the text at the various places where these new references were included.

Reviewer 2 Report

Hyperthermia treatment in combination with other treatment modalities like radiotherapy is extremely relevant topic. According to in vivo results and metaanalysis of clinical trials, the mechanism of action is still unknown and it's needs to be better understood to provide more benefits for patients. In this context the role of hypoxia can be crucial. 

The main problem with current paper is not complete bibliography especially about hypoxia. In lines 295-6 authors are showing very limited knowledge about current dose painting techniques. The papers from Halpern's Lab (Chicago University) should be cited where they showed (in Red Journal Paper, 2018) successful story about dose painting in mice!

Additionally, I think that good scheme about hyperthermia mechanism should be done to organize presented data. Right now it is quite chaotic why and when hyperthermia can induce the specific response. That is why it will be nice to present the effects on one graph with indirect and direct hyperthermia influence on cells, vasculature and immune system. Table 1 is still not an answer because is also not clear. 

During data analysis authors are trying to pay attention to very important hyperthermia conditions like time of heating and obtained temperature. In my opinion the source of hyperthermia treatment should be analyze too. 

Next, in figure 4 the information about general tumor hypoxia should be placed. 

Author Response

Reviewer 2

Comments and Suggestions for Authors

1. Hyperthermia treatment in combination with other treatment modalities like radiotherapy is extremely relevant topic. According to in vivo results and metaanalysis of clinical trials, the mechanism of action is still unknown and it's needs to be better understood to provide more benefits for patients. In this context the role of hypoxia can be crucial. 

The main problem with current paper is not complete bibliography especially about hypoxia. In lines 295-6 authors are showing very limited knowledge about current dose painting techniques. The papers from Halpern's Lab (Chicago University) should be cited where they showed (in Red Journal Paper, 2018) successful story about dose painting in mice!

Response: When our review was first proposed we were asked by the editor to limit our general comments about hypoxia, because another review by Prof. Bristow specifically focused on this issue. Furthermore, hypoxia has been the subject of other reviews. Having said that we did include minor comments about hypoxia that were necessary to understand the role of hyperthermia, especially regarding other techniques that have been used. For some of these we were rather brief and dose painting techniques probably too brief. To rectify that we have included the suggested reference (number 89) and some additional comments in the text (page 8, lines 308-313).

2. Additionally, I think that good scheme about hyperthermia mechanism should be done to organize presented data. Right now it is quite chaotic why and when hyperthermia can induce the specific response. That is why it will be nice to present the effects on one graph with indirect and direct hyperthermia influence on cells, vasculature and immune system. Table 1 is still not an answer because is also not clear. 

Response: This is an excellent suggestion by the reviewer and to address it we have now added a new extra figure (figure 5) which clearly summarizes the roles of hypoxia and hyperthermia. The text on page 11 (lines 460-469) has also been modified to include comments relevant to that figure.

3. During data analysis authors are trying to pay attention to very important hyperthermia conditions like time of heating and obtained temperature. In my opinion the source of hyperthermia treatment should be analyze too. 

Response: The source of hyperthermia treatment has no relevance to the tumor microenvironment issue, so does not play a role in the potential of hyperthermia to target hypoxia. Thus, we did not include any discussion of the source of hyperthermia treatment. However, we did included references (numbers 10-13, page 2, lines 80-84) that discuss this particular issue.

4. Next, in figure 4 the information about general tumor hypoxia should be placed. 

Response: We are unclear as to what exactly the reviewer is referring to here. Figure 4 is a meta-analysis of clinical studies comparing radiation against radiation and hyperthermia, so there is no “information about general hypoxia” that we could think of that would be relevant here.